

# Assessing atmospheric dust modelling performance of WRF-Chem over the semi-arid and arid regions around the Mediterranean

Emmanouil Flaounas[1], Vassiliki Kotroni[1], Konstantinos Lagouvardos[1], Martina Klose[2], Cyrille Flamant[3], and Theodore M. Giannaros[1]

[1]National Observatory of Athens, Athens, Greece
[2]USDA-ARS Jornada Experimental Range, Las Cruces, NM, USA
[3]LATMOS/IPSL, UPMC Univ. Paris 06, Sorbonne Universités, UVSQ, CNRS, Paris, France

*Correspondence to*: Emmanouil Flaounas (flaounas@noa.gr)

**Abstract.** In this study we aim at optimizing the WRF-Chem model performance for the purpose of operational forecasting of dust transport over the eastern Mediterranean. For this reason, we compare the model output to observations in order to assess its capacity to realistically reproduce the aerosol optical depth (AOD), focusing on three key regions: North Africa, the Arabian Peninsula and the eastern Mediterranean. Three sets of four simulations each have been performed for the six-month period of spring and summer 2011. Each simulation set uses a different dust emission parametrisation and for each parametrisation, the dust emissions are multiplied with various coefficients in order to tune the model performance. Our approach is based on the model assessment across spatial and temporal scales by comparing its outputs to AOD observations from satellites and ground-based stations, as well as airborne measurements of aerosol extinction coefficients over the Sahara.

Tuning the model performance by applying a coefficient to dust emissions may reduce the model AOD bias over a region, but may increase it in other regions. Concerning dust transport over the eastern Mediterranean, the model was shown to realistically reproduce the major transport events, however failing to capture the regional background AOD. Model assessment over the entire domain and simulation period shows that the model presents temporal and spatial variability similar to observed AODs, regardless of the applied dust emission parametrisation. However, when focusing on specific regions, the model's skill may vary significantly. Further comparison of the model simulations to airborne measurements of the vertical profiles of extinction coefficients over North Africa suggests that the model may realistically reproduce the total atmospheric column AOD. Finally, we show that the inclusion of a finer dust mode (less than 1 μm) in the model presents the advantage of relaxing unrealistically large atmospheric dust loads and yet reproducing realistic AOD values.

## 1 Introduction

The geographical belt composed by North Africa and the Arabian Peninsula constitutes the largest desert in the world (Tsvetsinskaya et al, 2002). This region is a major dust source, emitting annually large loads into the atmosphere and thus having a global impact on climate and air quality (Huneeus et al, 2011). While both North Africa and the Arabian Peninsula emit remarkable amounts of particulate matter, the Sahara desert constitutes the worldwide main source of dust. In fact, the Arabian Peninsula emits about one fifth of the estimated dust uptakes over North Africa (Taichu et al, 2006). Dryan et al. (1991) showed that dust intrusions in the Eastern Mediterranean from the Arabian Peninsula have a short duration (of the





order of a day) and take place within shallow atmospheric layers of up to 2 km above sea level, while African dust intrusions persist longer (2-4 days of duration) and transport takes place at atmospheric layers over 3 km of altitude.

Climatologically, the emissions of dust over both regions are higher during spring and summer (Engelstaedter et al, 2006; Taichu et al, 2006). During this period, the atmospheric dynamics over North Africa, the Middle East and the eastern Mediterranean are strongly impacted by the monsoon system of West Africa and India (Flaounas et al, 2012; Tyrlis et al, 2014). The Indian monsoon onset establishes a low pressure system that extends from the Indian Subcontinent to the eastern Mediterranean. A quasi-constant descending cell of air masses is located over the eastern Mediterranean with pronounced impact on the surface wind circulation over the region (Tyrlis et al, 2014). Under these conditions dust storms are frequent over the Arabian Peninsula (Miller et al, 2008), while the Mediterranean climate and dust emissions are strongly affected by the West African monsoon and the Saharan heat low (Chauvin et al, 2010; Wang et al., 2015). Indeed, early summer is of particular interest for West African dust emissions. At the end of June, the monsoon propagates towards the north, displacing the ITD (Intertropical discontinuity, a near surface convergence zone between the monsoon and the Harmattan wind) to 20°N over the main source areas of dust (Prospero et al. 2002; Sultan et al., 2007; Klose et al., 2010; Gazeaux et al., 2011). In particular, Engelstaedter et al. (2006) showed that the West African monsoon onset plays a key role in the regional seasonal maximum of dust emissions. While uptakes of dust may occur due to local meteorological events such as dust devils, wind surges, turbulent mixing of low-level jets and cold pools associated with convective systems (Bou Karam et al., 2008; Knippertz and Todd, 2012; Klose and Shao, 2013), synoptic scale systems may transport dust away from the continent with a global impact (D'Almeida, 1986; Prospero, 1996; Moulin et al., 1997; Kaufman et al., 2005; Bristow et al., 2010; Bou Karam et al., 2010; Prospero et al., 2014; Flaounas et al., 2015).

Despite the importance of the African continent as a worldwide major dust source, the quantification of dust emissions is still an open question and strongly resides to numerical modelling. However, modelling dust uptake is a delicate issue, subject to a variety of uncertainties associated with the model's capacity to realistically reproduce the near surface meteorological conditions, the applied dust emission parametrisation, the model's vertical and horizontal resolutions, as well as the surface-related input datasets, such as erodible areas (e.g. Menut et al, 2007; Haustein et al, 2015; Teixeira et al, 2015). Indeed, the results of the analysis of an ensemble of 15 models showed that the potential dust emissions of North Africa vary significantly, ranging between 400 to 2200 Tg per year (Huneeus et al., 2011).

Accurate forecasts of dust emission and transport are also a societal demand worldwide as they pertain to many health and economic issues, such as air quality. Ambient air pollution is now the world's largest single environmental health risk, causing 3.7 million premature deaths worldwide every year (World Health Organization, 2014; Lelieveld et al., 2015). Forecasting dust uptake and transport requires adequate parametrisations, input fields and tuning techniques in order for results to best match observations (Basart et al., 2012; Benedetti et al., 2014; Sessions et al., 2015). For instance, Flaounas et al., (2009) showed that the realistic simulation of a pollution episode in southern France depended strongly on the explicitly resolved dust emissions over North Africa. In another case study of a three-day dust event over the Bodélé depression in North Africa, Todd et al. (2008) showed that the simulated dust-related fields (such as dust flux and



concentration) from five models differed by at least an order of magnitude. The meteorological conditions were realistically reproduced by all five models, suggesting that uncertainties were mostly related to the dust emission parametrisations and/or corresponding land-surface input data.

In this study, we test the sensitivity of the Weather Research and Forecasting model with chemistry (WRF-Chem) Version 3.6.1 (Grell et al., 2005) to the dust emission parametrisation through the comparison of modelled and observed atmospheric optical depth (AOD) over a large region that includes North Africa, the Arabian Peninsula and the eastern Mediterranean basin. Our objective is twofold. First, we assess the model performance in key regions of dust emissions. Our second objective is to establish an empirically tuned dust forecasting model for the effective forecast of dust transport over the

eastern Mediterranean, an area to which dust is frequently transported during spring and summer.

    Our study concentrates to a six-month period when dust transport over the Mediterranean is expected to be high, from spring to summer of 2011. Summer 2011 was included in the evaluation period in order to benefit from aircraft measurements of aerosol extinction coefficient profiles that were acquired over the Sahara during the Fennec campaign

(Ryder et al., 2015). We performed three sets of simulations with each set using a different dust emission scheme. For every dust emission scheme we applied different tuning coefficients to the surface dust emission flux (a total of 12 simulations) in order to achieve a realistic representation of the spatio-temporal variability of the AOD as observed by satellites and ground-based AERONET stations, as well as aircraft extinction coefficient measurements. The WRF-Chem model has been previously used in several studies to investigate dust storms and dust interactions with atmospheric thermodynamics and

radiation using different dust emission parametrisations (e.g. Zhao et al., 2010; Smoydzin et al., 2012; Kalenderski et al., 2013). Su and Fung (2015) used WRF-Chem to assess its performance to simulate dust concentrations over East Asia using two different dust emission parametrisations. Their results showed significant differences in the WRF-Chem performance when different dust uptake parametrisations were applied. To the best of the authors' knowledge, this is the first comprehensive study on evaluating the model's performance with a focus on dust emissions over the area of North Africa,

the Arabian Peninsula, and the eastern Mediterranean.

## 2 Simulation set-up, observations and methods

### 2.1 Model configuration and sensitivity tests

    The WRF-Chem model was operated on the domain shown in Fig. 1 at a standard longitude - latitude projection with a horizontal resolution of 0.22° and 0.19°, respectively (on the order of ~22km). The domain is composed by 424x250 grid points and 40 vertical levels. All simulations have been performed for the period of 21 February 2011 to 31 August 2011. The model is initialized with zero dust concentrations. The one-week period from February 21 to February 28 has been used

as a spin-up period for building dust concentrations within the domain and has not been taken into account for the model assessment. The model was forced into its initial and boundary conditions by the ERA-Interim (ERA-I) reanalysis of the European Center for Medium-Range Weather Forecasts (Dee et al., 2011). Boundary conditions and sea surface temperature were updated every six hours.





The WRF model has been previously shown to realistically simulate the West African monsoon and heat low dynamics during spring and summer (Flaounas et al., 2011; Klein et al., 2015). Here, we use the Grell 3D ensemble scheme for convection (Grell and Devenyi, 2002), the WRF single moment five microphysics scheme (Hong et al., 2004) and the Yonsei University planetary boundary layer parametrization (Hong et al., 2006). In order to achieve a realistic representation

of the meteorological conditions as well as to reduce uncertainty in the atmospheric circulation due to the model internal variability, we nudged wind, temperature and water vapour at each grid point to the ERA-I reanalysis, except within the boundary layer. The grid nudging coefficient we used is $6\times10^{-4}$ s$^{-1}$. The decision to nudge the model towards the driving reanalyses was based on a comparison of 10-meter wind speed from both, the nudged simulations and a simulation where no nudging was applied, with SYNOP (surface synoptic) observations. Comparison results showed that nudging clearly

improves the model 10-m wind speed. This is particularly important for dust emissions, since the near surface wind speed has been reported to be among the dominant factors in WRF-Chem model parametrisations (Zhao et al., 2010). In particular, it was found that the employment of nudging reduces the model 10-meter wind speed absolute bias over North Africa by approximately 35%, while it also allows for a better subjective agreement between the observed and modelled synoptic-scale patterns associated with dust transport. Grid nudging has been previously shown to contribute to the realistic

reproduction of the atmospheric circulation during a severe dust event over India (Kumar et al, 2014).

The chemistry component of the WRF model is used in dust-only mode, where the model takes into account dust uptakes from the soil as the only source of particulate matter, and transports it as a passive tracer within the simulation domain, treating explicitly gravitational settling, and vertical mixing. Consequently, all simulations present identical meteorological

conditions and atmospheric circulations, i.e. unaffected by dust direct or indirect effects. Three dust emission schemes are considered which output dust emissions for five size bins with effective radii of 0.73, 1.2, 2.4, 4.8 and 8 μm. Namely: (a)The first scheme is that developed by Gillette and Passi (1988), which is incorporated in WRF-Chem within the GOCART model (Ginoux et al.,2001). In the scheme (G01 in the following), the surface dust emission scales with the fourth power of wind speed as soon as wind speed exceeds a threshold value. This threshold value is a modified version of

the relationships obtained by Bagnold (1941) and Belly (1964). (b) The second dust emission scheme is the parametrisation developed by Marticorena and Bergametti (1995), incorporated in WRF-Chem in the Air Force Weather Agency (AFWA) dust module (AFWA hereafter). The scheme parametrises dust emission caused by saltation bombardment and the vertical dust emission flux is proportional to the horizontal saltation flux. The latter is obtained using a modification of the expression proposed by White (1979). The proportionality between dust emission and saltation flux was empirically related

to soil clay content by Marticorena and Bergametti (1995). (c) The third emission scheme is that developed by Shao (2004; S04, hereafter) in the University of Cologne (UoC) dust module package. The scheme accounts for the emission mechanisms of saltation bombardment, aggregates disintegration and relates dust emission to the volume removal of the saltating particles. In the scheme of Shao (2004), vertical dust emission flux is also proportional to horizontal saltation flux, but the proportionality depends on soil texture and soil plastic pressure. The scheme of Shao (2004) was originally

implemented to predict four size bins, but was modified to have size bins consistent with those used in other parametrisations. Required land-surface input data sets for the scheme are soil type, vegetation cover.





In their default setup, all schemes employ the erodible area defined by Ginoux et al. (2001), which is based on topography (Figure 1). With this variable, Ginoux et al. (2001) attempt to account for the variable amounts of sediment available in basins compared to hills and ridges. Thus the erodible area has values between 0 and 1. One important difference exists in how erodible area is used in the parametrisations. The schemes of G01 and AFWA use erodible area as scaling factor to

reduce dust emissions, i.e. the calculated dust emission flux at a particular grid point is multiplied by the erodible area at the same grid point. In contrast, the S04 scheme uses the erodible area only to define areas of potential dust emission, i.e. dust emissions are calculated if the erodible area is non-zero, but the fluxes are not scaled with the erodible area. The use of erodible area to define dust sources and their quality is independent of the parametrisations of dust emission in the different schemes but does significantly affect the distribution of modelled dust emission in all three schemes, as well as the

magnitude of modelled dust emissions in the case of G01 and AFWA.

For each dust emission scheme, we perform four simulations where the dust emissions are multiplied by four different coefficients in order to increase or decrease the dust fluxes in the atmosphere. Preliminary tests showed that a coefficient equal to 1 for MB95 and G01 resulted in disproportionally high AOD values over North Africa compared to the scheme of

Shao (2004). Consequently, we chose coefficients to be different for the four simulations using the S04 scheme. Table 1 presents a summary of the 12 performed simulations set-up.

### 2.2 Observations and comparison approach

To compare modeled AOD with observations, we use the MODIS AOD observations at 550 nm from the Terra and Aqua

satellites, corresponding to version 6 of daily gridded data in 1°x1° grid spacing in longitude and latitude, provided by the Goddard Earth Sciences Data and Information Services Center (MOD08 D3 and MYD08 D3, combined dark target and Deep Blue, giovanni.sci.gsfc.nasa.gov/giovanni). Aqua and Terra satellites provide worldwide daily observations, having a 2330 km swath and crossing the equator at 1:30 pm and 10:30 am local time, respectively. Retrieval of MODIS aerosol data is performed by different algorithms (e.g. Hsu et al., 2004; Remer et al., 2005) according to the underlying surface type. The

accuracy of the AOD retrievals has been evaluated both on a global and regional scale, against AERONET sun photometer measurements (e.g. Levy et al., 2010; Sayer et al., 2013). From the MODIS database, we have also used measurements of Ångström Exponent (AE) over land (470−660 nm; MOD/MYD 08_D3_051), over ocean (550−865 nm; MOD/MYD 08_D3_051) and over deserts (412-470 nm; MOD/MYD 08_D3_6), as well as the absorption Aerosol Index (AI), taken from OMI-Aura (Ozone Monitoring Instrument) measurements (Torres et al., 2007). Following the same approach as in

Flaounas et al. (2015) the MODIS AOD dataset was filtered so that model evaluation is performed only for grid points and days that dust is present. For this reason, we took into account only AOD values when AE is lower than 0.7 and AI is greater than 1. Finally, we also use ground observations of AOD, taken by the aerosol robotic network (AERONET, Holben et al, 1998). In contrast to satellite observations, AERONET observations offer the advantage of continuous, high-temporal resolution measurements in the daytime over a given location where satellite coverage might not be always available.

Figure 1 shows the dust fraction of erodible surface at each grid point, as taken into account by the WRF-Chem simulations. As expected, the major sources of dust are located in North Africa and in the Arabian Peninsula. We focus on these regions



in order to validate the modelled dust emissions. A second focus is on the eastern Mediterranean in order to validate the model capacity in realistically reproducing the dust transport over this region. These three subregions are depicted by boxes in Fig. 1. Six AERONET stations have been chosen so that their locations are representative of the sub-regions of interest and their observations are available during the simulation period (Fig. 1).

Finally, airborne measurements of the lidar-derived extinction coefficient acquired over the western Sahara during the Fennec campaign are used to evaluate the vertical profiles of modelled dust. During Fennec campaign, the SAFIRE (Service des Avions Français Instrumentés pour la Recherche en Environnement) Falcon 20 was equipped with the LEANDRE Nouvelle Génération (LNG) backscatter lidar (Bruneau et al., 2015). The profiles of atmospheric extinction coefficient at 532

10 nm were retrieved using a standard lidar inversion method that employs a backscatter-to-extinction ratio of $0.0205$ sr$^{-1}$ (see Schepanski et al., 2013, for details). At this wavelength, the lidar signal is mostly sensitive to aerosols with radii ranging from 0.1 to 5$\mu$m, and hence to dust aerosols. Furthermore, over the African continent, close to the sources, desert dust particles are generally considered to be hydrophobic (e.g. Fan et al., 2004). Therefore, extinction associated with desert dust is generally considered to be a good proxy for dust concentration in the atmosphere. The retrievals have an estimated uncer-

15 tainty of 15%, a resolution of 2 km in the horizontal and 15 m in the vertical. Lidar-derived extinction coefficient profiles were averaged over 30 min (~350 km) along levelled legs performed by the Falcon 20 during 5 flights on 14, 15, 20, 21 and 22 June (see Ryder et al., 2015 for flight tracks). This was done to extract the main characteristics of the dust layers over the Sahara (vertical extent, magnitude of extinction) in an integrative approach more adapted to a comparison with model outputs which generally do not reproduce the high-spatial variability observed with lidars. The locations of the averaged verti-

20 cal profiles are shown as black dots in Fig. 1. The lidar-derived extinction coefficient profiles are compared to their simulated counterparts averaged over the same leg and extracted at the model output time closest to the time when the lidar profiles were acquired.

**3 Comparison of simulation results to observations**

**3.1 Model assessment in the simulation domain**

The seasons of spring and summer are expected to present the highest dust emission activity in the broader region including North Africa, Middle East and the Mediterranean (Moulin et al., 1998). Figure 2 shows the average dust AOD as retrieved by MODIS for the whole six month period of spring and summer 2011. Over North Africa, the higher AOD values are

30 observed along the 15°N latitudinal belt, at the climatological location of the inter-tropical discontinuity frontal area between the monsoon and the Harmattan wind. The higher AODs are observed downstream of the Bodélé depression. High dust concentrations are also located over the northern part of the Arabian Peninsula and are related to the Shamal winds continuously blowing over dust sources linked to the alluvial plains of Syria, Irak and western Iran. Large AOD values are also observed to be associated with emissions from the Aral Sea sediment basin, east of the Caspian Sea. The mean AOD

35 values of spring and summer are dramatically lower over the Mediterranean region where dust sources are limited.

The AOD differences between the WRF-Chem simulations and the MODIS estimations (as shown in Fig. 2) are presented





in Fig. 3. Differences correspond to the AOD six-month averages, taking into account only the days and grid points when MODIS provides measurements. As expected, in all simulations, the AOD bias varies over the whole region as a function of the dust flux coefficient. In Sim_G01-1 and Sim_G01-0.75 (Figs 3a and 3d), the AOD is largely overestimated over North Africa while when applying a coefficient of 0.5, the model seems to be in better agreement with the MODIS observations (Fig. 3g). On the other hand, the modelled AOD over the Mediterranean Sea seems to be closer to the observations in Sim_G01-1 and Sim_G01-0.75, while the model overestimates AOD over North Africa. In the Arabian Peninsula, Sim_G01-1 and Sim_G01-0.75 tend to overestimate AOD over the region's southeastern part, compared to the AOD over the northern side. This is consistent with the higher fraction of erodible surface in the south of the Arabian Peninsula, as shown in Fig. 1. Sim_G01-0.5 also appears to produce the most realistic AODs in that region. It is noteworthy that all the G01 simulations underestimate the AOD in the vicinity of the Euphrates and Tigris rivers basin.

Similar results are obtained using the AFWA and S04 schemes. Over North Africa, the AOD is overestimated for the larger tuning coefficients (Figs 3b and 3c), with a smaller bias over the Mediterranean. Sim_S04-1.5 appears to show the smallest bias of all S04 simulations over North Africa. In fact, the S04 simulations tend to present a large overestimation of AOD over three hot spots of dust emissions located in southern Iran, close to the Sistan region, in the northern part of the horn of Africa and in the eastern part of central Africa. These are areas which have a small fraction of erodible surface (compare Fig. 1), thus emissions produced with the G01 and AFWA schemes are already significantly reduced through multiplication with this fraction (Section 2.1). It cannot be ruled out that similar overestimations would occur without this second scaling. Overall, the lower tuning coefficients provide a general underestimation of AOD over the whole simulation domain, regardless the dust emission parametrization (Figs 3j, 3k 3l).

In order to quantify the WRF-Chem model skill in reproducing the six-month average AOD in all simulations, Fig. 4 presents the spatial Taylor diagram (Taylor, 2001) that compares MODIS observations (as presented in Fig. 2) to the simulation outputs. In the Taylor diagrams, the centred root mean square error (RMSE, in abscissa) provides a measure of the model total AOD differences from the observations within the entire domain, while the standard deviation (in ordinate) and correlation provide a measure of the models skill to reproduce the AOD spatial variability. Figure 4 shows that all simulations present a high correlation coefficient of the order of 0.75 to 0.8. This suggests a rather good skill of the WRF-Chem model in realistically reproducing the seasonal spatial variability of dust concentrations, regardless of which dust emission parametrisation is used. On the other hand, the RMSEs and standard deviations strongly depend on the applied tuning coefficients. In fact, Sim_G01-0.25, Sim_AFWA-0.25 and Sim_S04-0.5 seem to present standard deviations which are closer to MODIS, as well as the lowest RMSE. Although these three simulations underestimate the AOD compared to MODIS (see Figs 3j, 3k and 3l), their overall bias – averaged over the whole domain - is smaller than in the simulations using larger tuning coefficients as for instance Sim_AFWA-0.75, Sim_G01-0.75 and Sim_S04-1.5 (Fig. 3d, e, f, respectively). Small and moderate tuning coefficients limit the simulated hot-spots of high dust concentrations and thus the model standard deviation is closer to the observations. When comparing the simulations in their standard set-up (tuning coefficient equals to 1), the S04 scheme shows the smallest standard deviation and RMSE, followed by AFWA and G01.





### 3.2 Model assessment on regional scale

In order to evaluate the model skill in reproducing the AOD on regional scales, we focus on three sub-domains, depicted by boxes in Fig. 1a. For each simulation, Figure 5 shows the average absolute bias of modelled AOD with respect to MODIS derived AOD within each sub-domain and for the whole six-month simulation period. For each dust emission parametrisation, there is a coefficient that corresponds to a minimum absolute bias. As discussed in the previous section, all three simulation sets provide smaller biases over the Eastern Mediterranean domain when the tuning coefficients are large (Fig. 5c). On the other hand, smaller tuning coefficients seem to be more adequate for the North African domain. Indeed, Sim_G01-0.5 and Sim_AFWA-0.5 result in smaller biases for the African domain (Fig. 5a), while Sim_G01-1 and Sim_AFWA-1 tend to produce smaller biases for the Eastern Mediterranean. Sim_S04-1.5 achieves a minimum absolute bias over the North African domain (Fig. 5a), while Sim_S04-2 yields the minimum absolute bias for the Eastern Mediterranean domain (Fig. 5c). Regardless the dust emission parametrisation, the North African domain and the Arabian Peninsula do not share the same tuning coefficients for minimizing absolute errors despite that they are both regions with major dust sources. Indeed, Fig. 5b shows that larger tuning coefficients in G01 and AFWA (Sim_G01-0.75 and Sim-AFWA-1) tend to reproduce smaller biases in the Arabian Peninsula. There is an opposite behaviour of the simulation results obtained with S04. In fact, Sim_S04-1.5 produces a smaller bias for North Africa, while Sim_S04-1 (no tuning) yields a better performance for the Arabian Peninsula. Such a different behaviour between the schemes might be attributed to the different treatment of the potential dust source areas. Overall, the use of coefficients in order to tune the modelled dust emissions is shown to reduce or increase the model absolute bias of AOD over the chosen regions of interest, namely North Africa, the Arabian Peninsula and the Eastern Mediterranean. The optimal coefficient to minimize the regional AOD absolute bias is not the same for all regions.

To gain further insight on the capacity of WRF-Chem to reproduce the regional AOD, Fig. 6 shows time series of the daily evolution of AOD from WRF-Chem and MODIS, averaged over each of the three domains. For Africa, both model and MODIS show a strong overall variation in the domain averaged AOD, with few distinct peaks during the investigation period. All simulations qualitatively capture the timing of most periods with increased AOD, however, the double peak in late June and early July is not well reproduced by the parametrisations. A similar result is obtained for the Arabian peninsula domain shown in Fig. 6b, except that Sims-S04 strongly overestimate dust emissions starting from July onward. For the Eastern Mediterranean, Fig. 6c shows that the MODIS AOD observations present an average AOD background value of the order of 0.25, while several peaks are representative of major dust transport events (as for instance on 1 May 2011; Fig. 6c). Since the atmospheric circulation is identical in all simulations and nudged to the ERA-I reanalysis, the model realistically captures the time of the dust transport events, as reflected by the high correlation coefficient of about 0.7 for all simulations (Fig. 7c). On the other hand, if no dust transport takes place (as for instance during the second half of June 2011 in Fig. 6c) the WRF-Chem model AOD values are close to zero (Fig. 6c). Consequently, regardless the dust emission scheme, the model fails to realistically reproduce the background dust concentration over the Mediterranean. It is thus plausible to suggest that if no major dust transport event takes place in the region, the model excessively removes dust from the atmosphere over the Mediterranean and/or that other aerosol sources are not captured by the model.





Figure 7 shows Taylor diagrams that statistically assess the model and observed AOD time series, as shown in Fig. 6. In accordance with Fig. 4, Fig. 7 shows that different coefficients result in an increase or decrease of the average modelled AOD, without having a noticeable effect on the correlation. For the North African domain, Sims_AFWA show slightly better correlations compared to Sims_G01 and Sims_S04, suggesting that the simulations using AFWA applies better to

North Africa for the given model set-up (i.e. for the given domain, resolution etc.). Correlation coefficients are also slightly higher for Sims_AFWA for the Arabian Peninsula than for the two other simulation sets.

### 3.3  Model assessment at the local scale

AOD observations acquired from AERONET stations allow to assess the skills of WRF-Chem in reproducing the AOD on

local scale. Figure 8 shows the model simulated time series of AOD, interpolated at the locations of the six AERONET stations shown in Fig. 1. For the statistical assessment of WRF-Chem at the locations of the AERONET stations, we show the Taylor diagrams corresponding to the time series of Fig. 8, in Fig. 9. In North Africa, all simulations capture the increase of AOD in Zouerate after 15 June (Fig. 8a), i.e. during the period of installation of the Saharan heat low (Todd et al., 2013) over the central Sahara after the African monsoon onset took place (Cornforth et al., 2012). All simulations show equal

correlation coefficients of about 0.7 regardless of the tuning coefficients applied to the dust emissions (Fig. 9a). In agreement with the model results over the entire North African domain (Figs 6a and 7a), simulations with tuning coefficients smaller than 1 tend to result in smaller RMSEs and standard deviations which are close to the observations. The modelled AODs at Tamanrasset and Oujda are also in good agreement with the AERONET observations (Fig. 8b and 8c). While at Oujda  all simulations present equal correlation coefficients (Fig. 9c) as in Zouerate (but with a correlation of 0.4),

the correlations at Tamanrasset depend on the dust emission parameterization (Fig. 9b). This is due to the fact that in Tamanrasset, dust-related AODs depend on both long-range transport from remote North and East Africa sources and local emissions (Cuesta et al., 2008). Simulations using G01 show larger correlations than the simulations using AFWA and S04, suggesting a more realistic daily variability of dust concentrations over this site.

At the Solar Village in the Arabian Peninsula, larger correlation coefficients (~0.6) are obtained for the simulations using G01. All simulations tend to underestimate the standard deviation and have RMSEs of more than 0.3 (Fig. 9d). Indeed, all simulations seem to underestimate the average AOD during all the six-month period (Fig. 8d). In consistency with the model results at Tamanrasset, the G01 simulations at the Solar Village present better correlations than both AFWA and S04. It is rather difficult to explain the reasons for this consistency in the model performance. However, it seems that in both

cases convection may be largely connected with dust outbreaks (Guirado et al, 2014; Houssos et al, 2015).

For the Mediterranean, we compare the AERONET station observations in Crete and Lampedusa with the model results. All simulations were able to reproduce the major dust transport events corresponding to the peaks in AOD (Fig. 8e and 8f). This is also reflected by correlation coefficients of the order of 0.6 (Crete) and 0.7 (Lampedusa) for all simulations, as shown in

Fig. 9f and Fig. 9e, respectively. Despite the high correlation coefficients, all simulations underestimate the background dust concentration in these stations. Indeed, all simulations show AOD values close to zero except when dust transport events take place (Figs 8e and 8f). On the other hand, AERONET observations from both Crete and Lampedusa present values



close to 0.25, consistent with the MODIS average regional AOD values shown in Fig. 6c.

### 3.4 Model assessment of the vertical distribution of dust over the Sahara

To gain further insight in the model's capacity to reproduce dust uptake and transport, Fig. 10 shows the vertical profiles of
the lidar-derived extinction coefficients from 5 flights between 14 and 22 June 2011, as well as the corresponding values
obtained with WRF-Chem. Airborne measurements have been taken over Northern Mauritania and Northern Mali, in the
vicinity of major dust sources (Fig. 1). We present only Sim_G01-0.5, Sim_AFWA-0.5 and Sim_S04-1.5 which have the
smallest biases over these locations among all simulations (Fig. 3). Results are highly variable depending on the flight. In
Fig. 10a all models seem to capture the vertical profile shape with a decrease of dust concentrations with increasing height,
and a sharp decrease in extinction around 5 km amsl, marking the top of the Saharan atmospheric boundary layer (SABL).
Nevertheless, the model seems to overestimate the total AOD due to excessive dust concentration throughout the
atmospheric column. The lidar-derived extinction profile acquired on 14 June is representative of the low dust concentration
over Northern Mauritania and Northern Mali when the Western Sahara was under the influence of cold air masses from the
Atlantic (Todd et al., 2013). In subsequent flights, lidar profiles were acquired while the Western Sahara was under the
influence of the approaching Saharan heat low as well as strong low-level northeasterly wind surges from the Mediterranean
(Todd et al., 2013). The wind surges were responsible for enhanced emissions in the Western Sahara and for the large AODs
observed in Zouerate during the second half of June (Fig. 8a). As for Fig. 10a, the observations in Fig. 10b show that dust
concentrations tend to decrease with height, large extinction coefficient values being observed near the surface as the result
of dust emissions. During the Saharan heat low phase, i.e. on 15, 20, 21 and 22 June, lidar data evidence essentially a two-
layer structure in the SABL, with a deep well mixed upper layer (above 1-1.5 km amsl, Fig. 10b-f) and a layer of enhanced
extinction in the lower 1-1.5 km amsl corresponding to the freshly emitted dust in the source regions. The model fails to
capture the observed high extinctions in the lower layer, but has a fairly good performance at reproducing the structure of
the SABL as well as the magnitude of the extinction coefficients derived from lidar. The extinctions in the upper part of the
SABL are associated with the long-range transport of dust from remote north and easterly sources, a process that is well
captured by the model. On the other hand, extinctions in the lower layers are related to small scale processes that are not
captured by the simulations owing to the relatively coarse mesh size of the model (i.e. 22 km). A striking example of that is
shown in Fig. 10d, where the low-level extinction values were observed to be the largest during the Fennec campaign. They
were caused by the cold-pool of a convective system having developed overnight over the Atlas Mountains, which then
propagated south-westward over the Sahara (Todd et al., 2013; Ryder et al., 2015; Chaboureau et al., 2016) and was
sampled by the lidar. The development of the convective system over the Atlas and the related cold-pool can only be
captured by convection permitting models as shown by Chaboureau et al. (2016) with mesh size on the order of 5 km or
less. Except maybe for the 21 June case, under the particular circumstances detailed above, the Sim_G01-0.5 simulation
always exhibits the largest extinction coefficients in the SABL. For most flights, the simulated extinction profiles were seen
to lay within the observed values if we account for the natural variability sampled by lidar along the Falcon 20 legs.

### 4 Discussion



### 4.1 On the relation between dust concentration and AOD

Overall, the results of the model comparison against observations showed that the modelled spatial and temporal variability in AOD is rather insensitive to the coefficient applied to the dust emissions. In fact, all simulations using the same dust emission scheme tend to present the same correlation coefficients when compared to observations, whether we consider local scales or the entire domain. In terms of the AOD level in the eastern Mediterranean, the model failed to reproduce the regional background value (of the order of 0.25). This was shown over the regional domain in Fig. 6, as well as in the local AERONET stations of Crete and Lampedusa (Fig. 8). However, the model showed good skill in capturing the dust transport events. Indeed, the modelled AOD time series over the eastern Mediterranean presented a large correlation coefficient of 0.7, when compared to the MODIS observations (Fig. 7c).

In this study we used extinction coefficients as a proxy for the vertical profile of dust concentrations. Comparing aircraft measurements of vertical profiles of extinction coefficients with the simulations, it was shown that even with different dust emission parametrisations, simulations tend to reproduce similar profiles, i.e. extinction coefficient profiles decreasing with increasing height. Nevertheless, differences are observed for simulations using different dust emission parametrisations. For instance in Fig. 10d, with respect to the simulation using AFWA, the simulation using G01 slightly overestimates the dust extinction coefficients above the altitude of 5 km and underestimates them below. The AOD is a convenient field for assessing chemistry transport models since simulations may be compared to observations from a network of ground stations and satellites. On the other hand, these observations might provide misleading results on the model performance. Indeed, due to compensating biases in the upper and lower part of the SABL (overestimation/underestimation of the extinction coefficients above/below 1.5 km), the AOD values derived from the simulated profiles are found to be realistic, especially for Sim_G01-0.5 and Sim_S04-1.5 during the second half of June, during the so-called Sahara heat low phase (Fig. 10b-e). These compensating biases were also highlighted by Chaboureau et al. (2016), even for higher resolution simulations performed with convection permitting models. Here we presented only five profiles of extinction coefficient, but averaged over several hundreds of kilometres along the flight legs to average observed outliers, which are thought to be representative of the model performance.

### 4.2 On the impact of dust bins selection to model performance

Our results derive from AOD observations compared to model outputs. However, the AOD reflects the dust load within the atmosphere and consequently it only provides indirect information on the dust concentration, especially in the lower atmospheric levels. Figure 11 shows the near-ground average dust concentration (i.e. the dust concentration at the first model level) during the whole six-month period for each simulation. The three simulations using the larger coefficients (Figs 11a, 11b and 11c) show average dust concentrations over North Africa which may exceed 1200 $\mu$g m$^{-3}$. No PM10 observations were available for performing a long-term direct comparison with the model simulations, however, the near ground modeled dust concentrations seem to be excessively overestimated by the model. Indeed, PM10 observations along the Sahel (along ~14°N) show values that are typically less than 100 $\mu$g m$^{-3}$ in spring and summer (Marticorena et al., 2010). In addition, comparing the order of magnitude of modeled dust concentrations with measurements at specific stations in the Mediterranean (Pey et al., 2013), results show that the model tends to produce dust concentrations that are an order of





magnitude larger than observations during episodes of dust transport over the Mediterranean (not shown). Consequently, relatively small coefficients (such as the ones used at Sim_G01-0.25, Sim_AFWA-0.25 and Sim_S04-0.5) seem to be more adequate for the proper representation of dust concentration over the African continent and for dust transport in the Mediterranean. On the other hand, our results in Fig. 3 suggest that these simulations lack of a realistic representation of

AOD within the whole simulation domain.

In order to achieve overall realistic values of AOD, the WRF model configurations assessed here produce extensively large dust surface concentrations. Consequently, there is a competitive effect on the model's performance on AOD and dust concentration reproduction. More realistic values of AOD would demand unrealistically high dust concentrations and vice

versa. A plausible explanation resides to the model configuration and the use of five particle-size bins. The effective radii of the dust particles considered here, mostly refer to coarse particles of more than 1μm. However, the dust aerosol extinction efficiency is expected to be maximum for dust particles of finer size, around 0.5μm. To investigate the potential of improving the WRF model performance on both AOD and the reproduction of realistic near ground concentrations of dust, we implemented eight dust sizes in WRF, following Basart et al. (2012). We performed an additional simulation, using the

dust emission parametrisation of G01 and eight dust-size bins with radii 0.15, 0.25, 0.45, 0.78, 1.3, 2.2, 3.8 and 7.1 μm, as well as a tuning coefficient of 0.25. Our simulation results on AOD bias and near ground dust concentration are shown in Fig. 12. Evidently, the use of 8 dust-size bins, produces average dust concentration of an order of magnitude lower compared to Sim_G01-0.25 (Fig. 12a versus Fig. 11j), however the AOD average bias is closer to the level of Sim_G01-0.5. Therefore, when using fine dust sizes with higher extinction efficiency but with weaker impact on the atmospheric dust

load, the model performance seems to acquire more realistic results for near ground dust concentration. However, a more detailed assessment must be performed using PM10 observations as a reference.

### 5 Summary and conclusion

In this study, we assessed the WRF-Chem model capacity to realistically reproduce the dust AOD over the broader region of North-Africa, the Middle East and the Mediterranean for the six-month period from spring to summer 2011. We performed three sets of simulations, each using a different dust emission parametrization. For each simulation set we multiplied different tuning coefficients to the parametrized dust emission fluxes, aiming at minimizing the model's AOD bias in different regions. Our approach resided in comparing the model results to AOD observations across different temporal and

spatial scales, using satellite, ground-based and airborne observations.

The meteorological conditions and atmospheric circulation were identical in all simulations. Therefore, all differences in AOD originated from the different dust emission parametrisations. When compared to AOD observations, the assessment of the simulations showed that regardless the coefficient used, the model produces similar correlation coefficients for the

simulations that use the same dust emission scheme. Consequently, tuning the emissions by a coefficient resulted only in reducing or increasing the AOD model bias. When considering regional or time averaged model outputs, all three different parametrisations that we tested seemed to present quasi-equal correlation coefficients with the observations. However, when



comparing model outputs to local stations, in four out of six stations (Tamanrraset, Lampedusa, Crete and Solar village ), the simulations using G01 and AFWA presented slightly larger correlation coefficients than S04. In its default implementation (i.e. using a tuning coefficient of 1), the simulation using S04 showed smaller RMSEs for four out of the six stations than those using G01 or AFWA. Comparing the model to airplane observations -and given its spatial resolution- the

5 model shows a fairly good skill in reproducing the vertical profiles of extinction coefficients over Northwest Africa.

The motivation of this study is to determine an optimal model set-up for a given domain in order to properly forecast dust concentrations over the eastern Mediterranean. Therefore, a simulation presenting the smallest bias and largest correlation coefficient would be the most adequate choice. However, our results show that there is no optimal model set-up that could

minimize bias simultaneously in all three regions of interest. Consequently, the simulations with low coefficients of the order of 0.5 seem to provide a reasonable trade-off choice in order to properly reproduce major dust transport events over the Eastern Mediterranean, as well as realistic levels of AOD over the desert belt of North Africa and the Arabian Peninsula.

In order to achieve a more realistic model performance in both AOD and near surface dust concentration, future work will

be concentrated to the climatology of dust transport over the Mediterranean by performing long term simulations and using eight aerosol size-bins. Additional sensitivity tests will be performed in order to test the model capacity in reproducing realistic dust transport events for finer model resolutions, aiming also at investigating the aerosols direct and indirect effect during extreme dust transport events over the Mediterranean.

**Acknowledgments**

This publication was supported by the European Union Seventh Framework Programme (FP7-REGPOT-2012-2013-1), in the framework of the project BEYOND, under Grant Agreement No. 316210 (BEYOND - Building Capacity for a Centre of Excellence for EO-based monitoring of Natural Disasters). The authors are grateful to NASA for providing the AERONET and MODIS datasets, as well as to the PIs and associated teams for the datasets maintenance and availability. Analyses and

25 visualizations used in this study were produced with the Giovanni online data system, developed and maintained by the NASA GES DISC. Airborne data were obtained during the FENNEC campaign using the Falcon 20 Environment Research Aircraft operated and managed by Service des Avions Français Instrumentés pour la Recherche en Environnement (SAFIRE, www.safire.fr), which is a joint entity of CNRS, Météo-France, and CNES. The Fennec-France project was funded the Agence Nationale de la Recherche (ANR 2010 BLAN 606 01), the Institut National des Sciences de l'Univers

(INSU/CNRS) through the LEFE program, the Centre National d'Etudes Spatiales (CNES) through the TOSCA program, and Météo-France.

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





**Tables**

| Simulation names | Description |
|---|---|
| Sim_G01- ## | Dust emissions after Ginoux et al. (2001)<br>## stands for the coefficient multiplying emissions: 1, 0.75, 0.5 and 0.25 |
| Sim_AFWA- ## | Dust emissions based on Marticorena and Bergametti (1995)<br>## stands for the coefficient multiplying emissions: 1, 0.75, 0.5 and 0.25 |
| Sim_S04- ## | Dust emissions after Shao (2004)<br>## stands for the coefficient multiplying emissions: 2, 1.5, 1 and 0.5 |

**Table 1** Simulations description

**Figures**

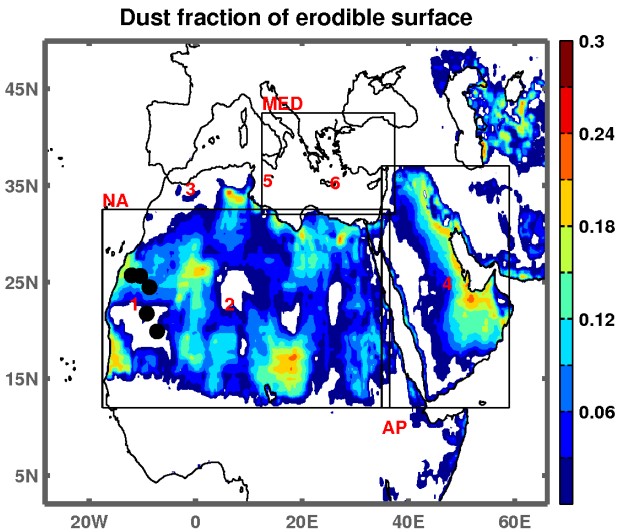

10     **Figure 1:** Fraction of erodible surface after Ginoux et al. (2001). Boxes depict the three sub-regions of North Africa (NA), Arabian Peninsula (AP) and the Eastern Mediterranean (MED). Numbers represent the locations of AERONET stations used in this study and black bullets show the locations of airplane retrievals of the vertical profiles of extinction coefficients. The AERONET stations are: (1) Zouerate, (2) Tamanrasset, (3) Oujda, (4) Solar Village, (5) Lampedusa, and (6) Crete.





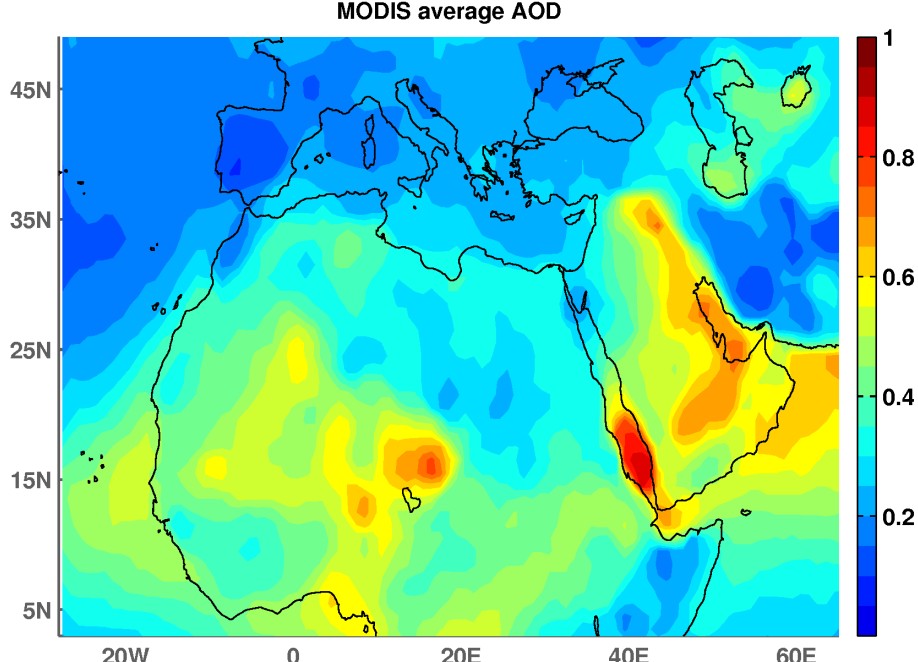

**Figure 2:** MODIS AOD observations within the simulation domain, averaged for the whole six month period, i.e. 1 March to 31 August 2011.





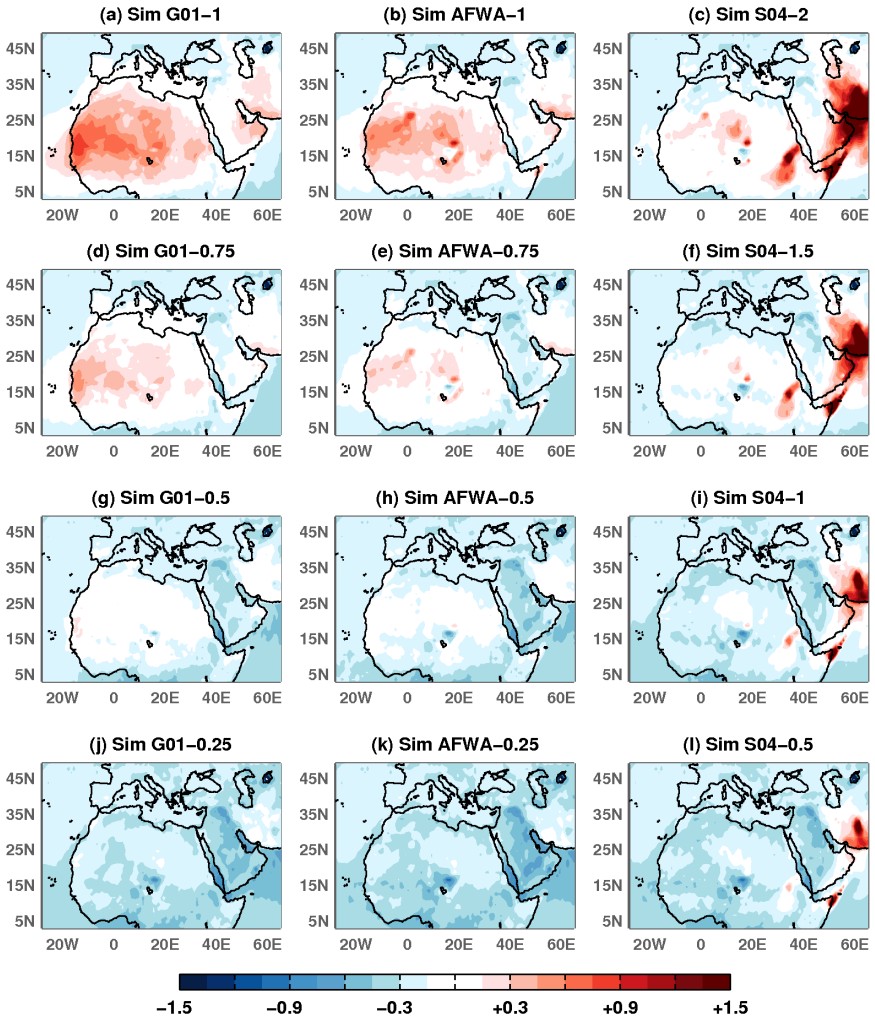

**Figure 3:** Differences between the modeled and observed AOD, averaged over the 6-month period. Note that different coefficients are applied for simulations using S04 compared to the ones using G01 and AFWA.





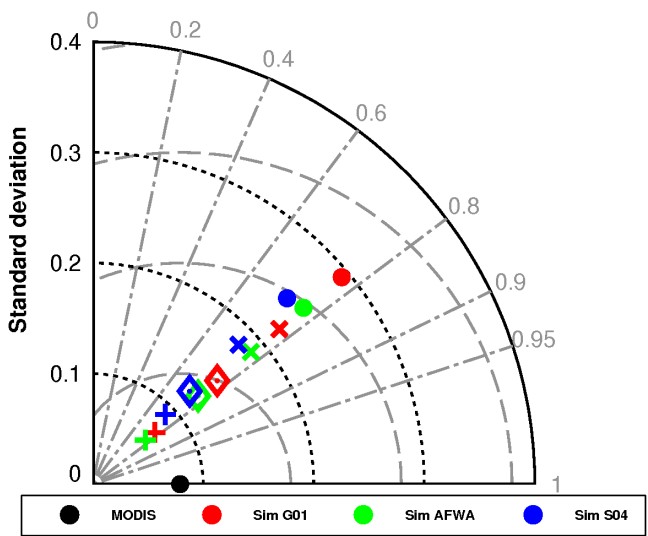

**Figure 4:** Taylor diagram comparing the six-month AOD average of all simulations with MODIS observations for the region illustrated in Fig. 2. Root mean square error lines (gray dashed circular lines) and standard deviations (*blacked dotted lines*) are plotted with an interval of 0.1, while correlation coefficients are shown by the *gray radii lines*. Symbols in red stand for Sims_G01, in green for Sims_AFWA and in blue for Sims_S04. The black dot stands for MODIS, dots (.) for Sim_G01-1, Sim_AFWA-1 and Sim_S04-2, (*X*) for Sim_G01-0.75, Sim_AFWA-0.75 and Sim_S04-1.5, Diamond (◊) for Sim_G01-0.5, Sim_AFWA-0.5 and Sim_S04-1, cross (+) for Sim_G01-0.25, Sim_AFWA-0.25 and Sim_S04-0.5.

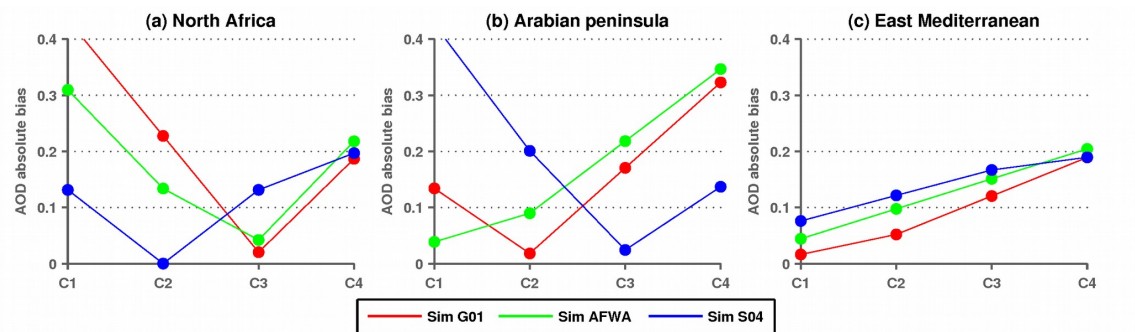

**Figure 5:** Average absolute bias between the simulations and MODIS observations for the whole six-month period and for the three sub-domains, depicted in Fig. 1. The x-axis values of C1, C2, C3 and C4 correspond to the coefficients applied for each simulation set. C1 equals 1, 1 and 2 for Sims_G01, Sims_AFWA and Sims_S04, respectively. C2 equals 0.75, 0.75 and 1.5, C3 equals 0.5, 0.5 and 1 and C4 equals 0.25, 0.25 and 0.5.





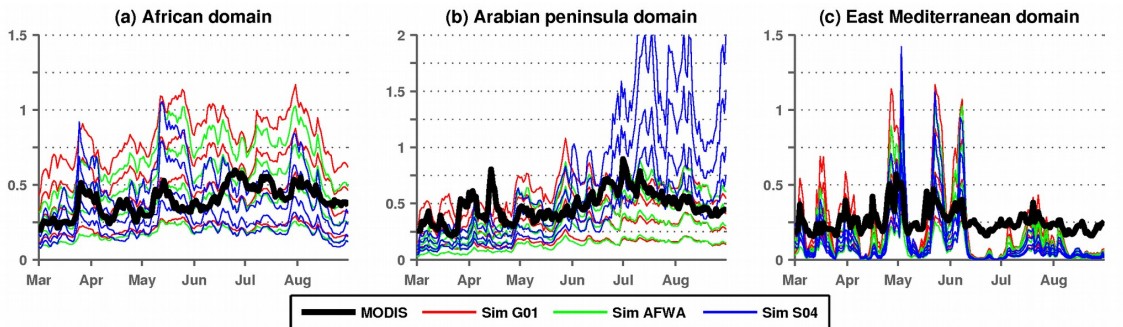

**Figure 6:** Time series of the daily averaged AOD for the simulations and MODIS for the whole six-month period, averaged over the three sub-domains depicted in Fig. 1.

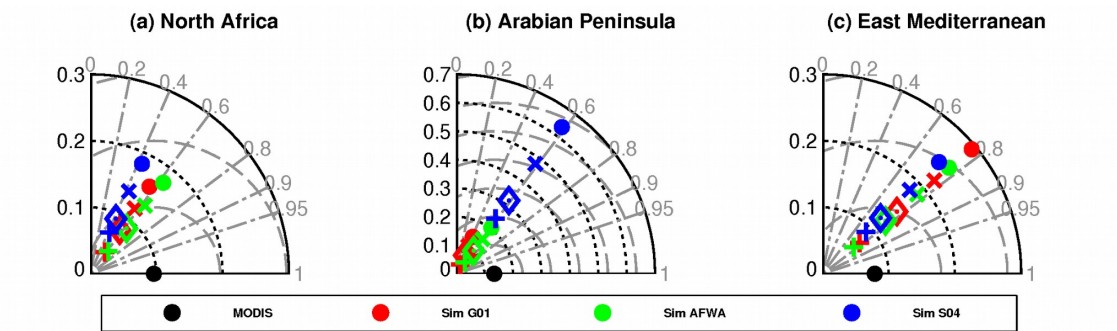

**Figure 7:** Taylor diagram comparing time series of AOD for all simulations to the MODIS observations as shown in Fig. 6. Root mean square error lines are plotted with a 0.1 interval. Symbol annotations are the same as in Fig. 4.





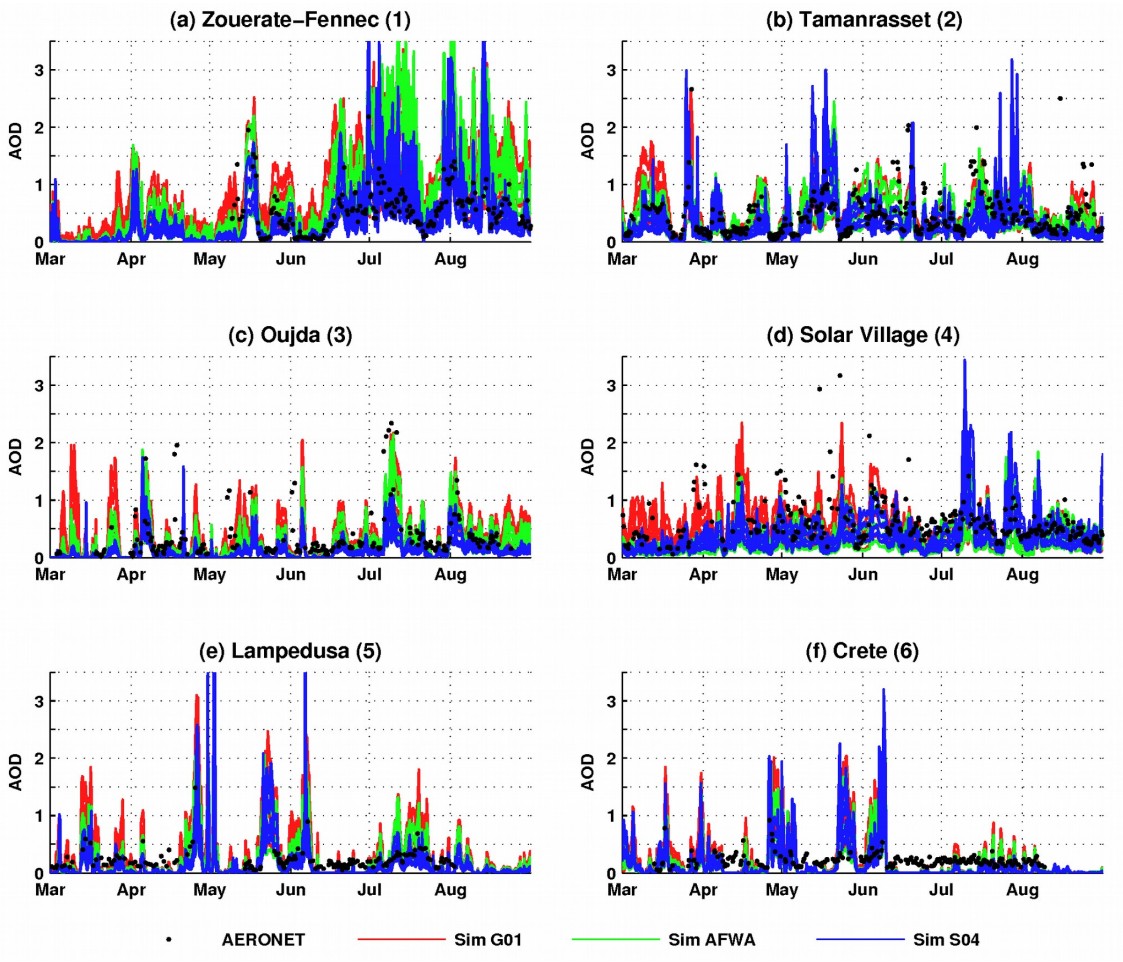

**Figure 8:** Time series of AOD for the simulations and AERONET observations during the whole six-month period. AERONET station locations are shown in Fig. 1.





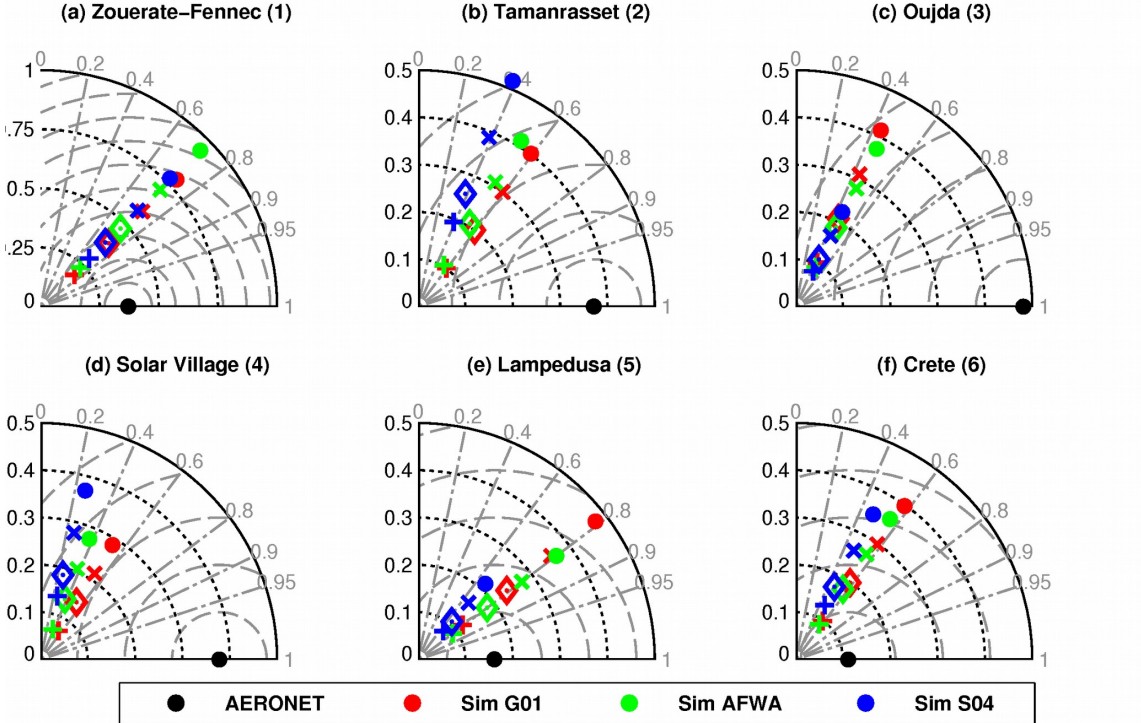

**Figure 9:** Taylor diagram comparing time series of AOD for all simulations to the AERONET station observations as shown in Fig. 8.

Root mean square error lines are plotted with a 0.1 interval. Symbol annotations are the same as in Fig. 4.




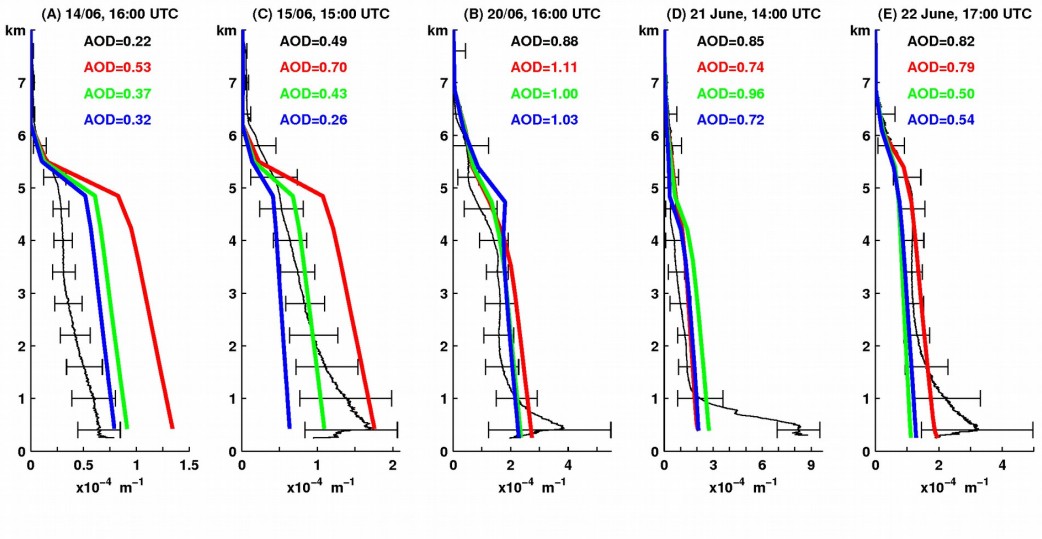

**Figure 10:** Extinction coefficient vertical profiles from the airborne lidar observations (black solid line) and from the WRF-Chem simulations (see legend for colors). The AOD values corresponding to the profiles are shown within the five panels. Error bar lengths equal twice the standard deviations of the lidar measurements at a given altitude.





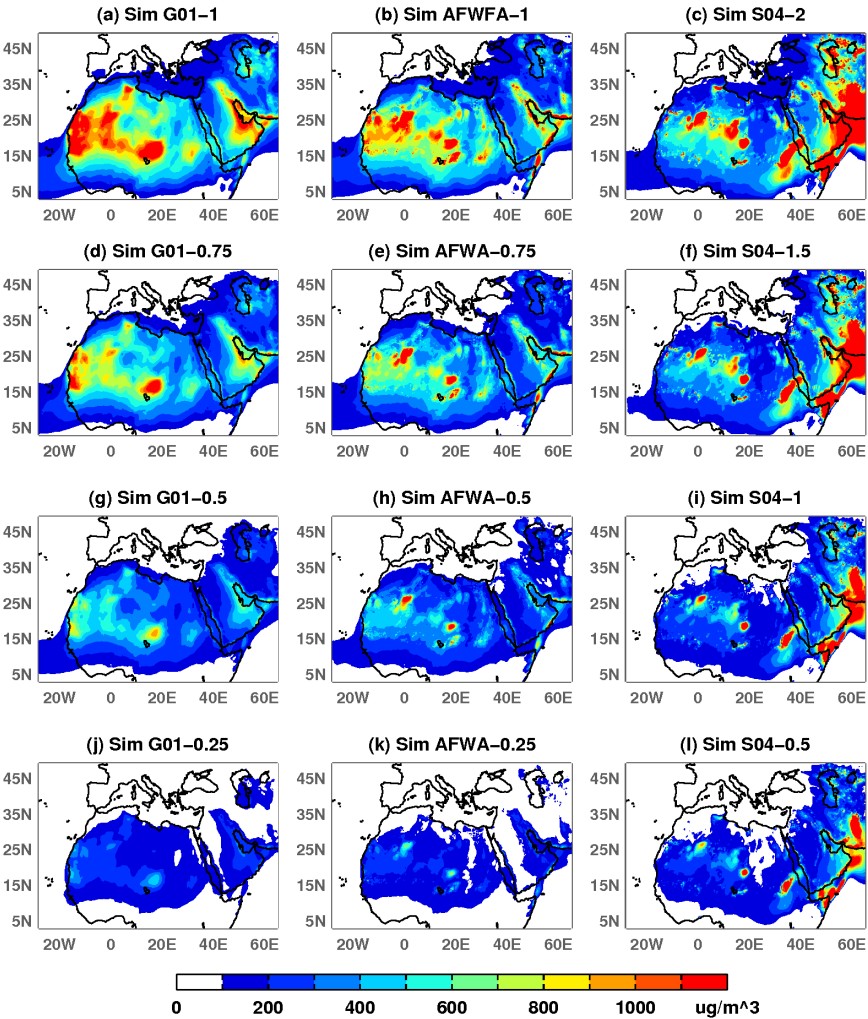

**Figure 11:** Near ground dust concentrations for all simulations, averaged over the 6-month period.





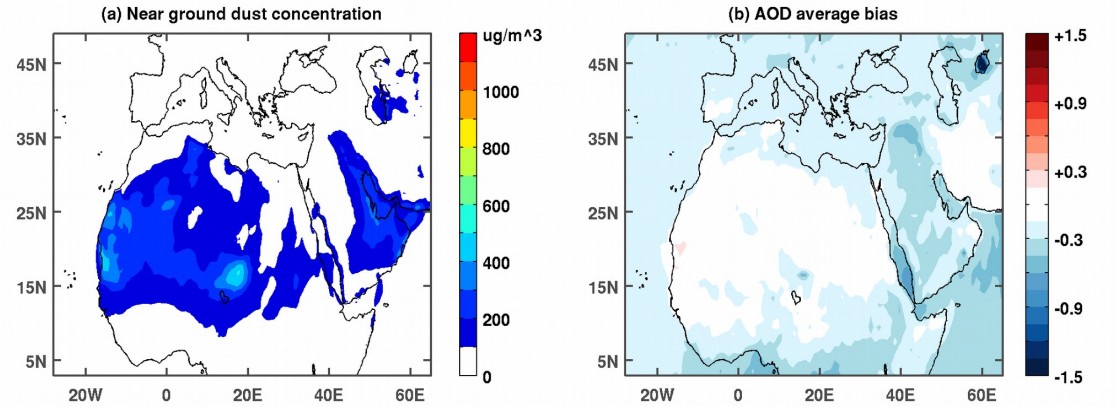

**Figure 12: A** Simulated near ground dust concentrations using G01 with a coefficient of 0.25 and eight dust-size bins, averaged over the 6-month period. **B** AOD differences between the simulation and the MODIS observations, averaged over the 6-month period.