# Peer review of "Assessing atmospheric dust modelling performance of WRF-Chem over the semi-arid and arid regions around the Mediterranean"

_Atmospheric Chemistry and Physics, 2016_

## Referee Comment (RC1) · Anonymous Referee #3 · 5 Jul 2016

**Review of the acp-2016-307 paper**

This paper reports on testing the performance of a regional dust-atmospheric modeling system. The study aims at optimizing the WRF-Chem model performance with added dust aerosol component in order to be capable to operationally forecast of dust transport over the eastern Mediterranean. The presented model is another one in the family of dust prognostic systems which development follows the interest of community to better predict dust process and its various impacts.

The authors successfully performed a series of tests to understand the performance of three used emission schemes, and to tune the model to achieve the optimal accuracy in different regions of the model domain. I recommend this paper to be accepted for publication after the authors consider suggestions and revisions as listed below:

Page 1 line 28:
*Tuning the model performance by applying a coefficient to dust emissions*
I agree this is the most straightforward way to vary the intensity of emissions and accept it as one of ways to tune the model. However, by this approach only a linear change of values every time everywhere is done. There are other possibilities as well such as e.g. modifying values of the threshold surface wind or friction velocity, aeolian surface roughness length, etc. Please discuss more this aspect and other possible ways for tunning.

Page 3 line 138:
*we nudged wind, temperature and water vapour at each grid point to the ERA-I reanalysis*
The authors claim that one of their objectives is ...to establish an empirically tuned dust forecasting model for the effective forecast of dust transport... By using nudging, operational features of the model could be contaminated. Once used, why nudging is not applied only to wind as the most critical parameter for emission? My general concern is that frequent nudging as applied in the experiment could affect the thermodynamic features of the atmosphere with unknown consequences. Please discuss possible impact of nudging to the operatibility of the model and eventual affecting the model thermodynamic balance.

General:
The presented extensive verification is certainly a good guidance how to select model setup based on more reliable emission options. However, since the authors' intention is to have a well tested model to be used for forecasting purposes, I strongly suggest that they select one of major dust storms during the considered experiment period and present a more close-up view so that a reader could get a better feeling on the model capability to successfully predict particular dust events.

---

## Referee Comment (RC3) · Anonymous Referee #2 · 11 Jul 2016

Review of "Assessing atmosphere dust modeling performance of WRF-Chem over the semi-arid and arid regions around the Mediterranean" by Flaounas et al. for publication in Atmospheric Chemistry and Physics

The paper presents simulations of the dust loading over three regions of interest (northern Africa, the Mediterranean, and the Middle East) for the six month period of spring and summer 2011. Simulations are made with the WRF-Chem model nudged by meteorology from the ERA-Interim reanalysis. The dust component of the chemistry module was run, and dust was carried as a passive tracer (i.e., no radiative or cloud interactions). Three different dust emissions schemes were considered in order to assess their relative performance at simulating observed dust distributions. Detailed comparisons were made to MODIS and AERONET AOD and aircraft lidar extinction profiles. Each of the three dust emission schemes were run with different scaling factors in order to investigate bias relative to observations. The general conclusion is that no one of the emission schemes tried can be optimally configured to provide the best performance over the three regions of interest. There is a secondary conclusion that suggests that simulated surface mass concentrations for the default configurations are excessively large, which can be somewhat ameliorated by including smaller sized dust particles in the simulation (which have lower mass but greater extinction efficiency).

My recommendation is that the paper needs major revision. I basically have two major criticisms to the work presented.

The first is that the analysis is far too simplistic as presented. Because you are using fixed meteorology and there is no feedback between the dust load and the atmosphere then there is really no point in performing 4 simulations for each of the emission schemes. Of course they have the same correlation coefficient. You are only running dust and no fancy microphysics and the processes simulated are linear (aren't they? I think they are in most models). The rescaling to different tuning coefficients could have been done a posteriori and essentially the same results derived. The analysis could be increased in complexity one notch by doing a regression to find the optimal tuning coefficient for each of the source schemes. This of course would not resolve the regional disparities. So maybe a next level of complexity would be to run the simulation turning on and off individual source regions (say, separate eastern from western North Africa an separate from Middle Eastern sources) and then you might find an optimal set of scaling coefficients for each region and each emission scheme (actually, this would require I think only 3 runs for each scheme. . .all sources, eastern Africa sources, and Middle East sources only, with the western African sources, for example, recovered by differencing the other simulations). This works because no feedback and processes are linear. This would yield some better estimate of the emissions needed in your model to make things work. (As an aside I see that the G01 and MB95 schemes yield

very similar results, and they differ from the S04. You don't really get into why this is the case, and that is perhaps interesting.)

The second is that the entire discussion of the dust particle size distribution Section 4.2 is too simplistic. Wouldn't the Fennec observations have some bearing on this discussion? Aren't there are in situ observations of dust mass to compare directly to? The problem I have with this entire line of analysis is that the authors never say how the dust particle size distribution is emitted at the source. For example, the Ginoux et al. 2001 prescription provides one example of an initial particle size distribution. Is that what is being used here? How about MB95, which does not provide explicitly the simulated particle size distribution at emission? And for S04? So some assumption is made about the initial particle size distribution, and again, couldn't the Fennec observations be used to evaluate that? What is not clear is why simply switching from 5 to 8 size bins reduces the simulated dust mass. Obviously you also changed the initial particle size distribution. How? Couldn't you have changed the distribution using the 5 bins to achieve a similar result? All that is described here is what the bin effective radius is, not what the size distribution actually looks like. There's nothing mysterious here about this: if you change the size distribution you change the mass extinction efficiency. Actually, that would be a useful thing to show, how that is different among the simulations.

Minor points:

Page 1: Line 25: Should read "however fails to capture..." Line 38: Should read "...the Arabian Peninsula has annual dust emission about one fifth as large as the emissions from North Africa..."

Page 2: Line 22: Instead of "resides to" use "relies on"

Page 4: Line 22: Why do you call the first scheme G01? You are not using the emission equation from Ginoux et al. 2001, but rather the equation from Gillette & Passi 1988. You are using the Ginoux source map, but you are using that after all for all the

simulations. As an aside, could you clarify what resolution of the Ginoux map you are using? Ginoux et al. 2001 describes a 1 degree source map, but he has separately provided a version of that map at 0.25 degree.

Page 5: The entire first paragraph is confusing to me. I just don't understand the distinction you are making here between S04 and the G01 and MB95 sources. Is S04 using some other source of sub-grid scale information prior to using the Ginoux map? Otherwise, why isn't this exactly the same as the other two? Fundamentally, don't you use the grid cell resolved winds with some flux equation, and then scale the resulting flux by the efficiency of the grid cell for emissions? Is there some difference in thinking of the Ginoux map as being an efficiency factor versus the fraction of the grid cell available for emissions? Unless there is some other source of sub-grid scale information I don't see what the difference is. Line 31: The OMI data is available nearly time-coincident with the Aqua overpass, but offset in time from the Terra overpass. How do you cope with that in driving this comparison of the model to MODIS data?

Page 7: Line 7: I don't see how G01-0.75 overestimates the Arabian peninsula AOD in Figure 3d. Line 9: How do you conclude that G01-0.5 produces the most realistic AOD over the Arabian peninsula when it looks more like G01-0.75 does (more white space)? Line 18: You refer to a second scaling. See my comment above about S04 scheme. I guess I'm confused about how the Ginoux map is being applied.

Page 9: Line 36: I find this conclusion about the background AOD confusing. I could argue that the simulated AOD is less than the background from AERONET at Crete, but it is not clearly the case for Lampedusa. But AERONET shows total column AOD, and you are only simulating the dust component. So I don't buy anything you are saying about the background AOD level. Am I missing something here? Do you have some reason to think that the AERONET AOD shown is due entirely to dust?

Page 10: Line 23: There is no analysis presented which supports the assertion that the higher altitude part of the dust AOD in the simulation results from long-range transport

as opposed emission processes. My experience, admittedly with global models, is that a deep boundary layer develops over Saharan Africa, so that dust may be mixed quite. So what is the PBL height in these simulations and along these profiles?

Figure 10: Use a consistent date labeling for each of sub figures (see title at top of each figure).

---

## Author Comment (AC1) · 3 Sep 2016

*Review of "Assessing atmosphere dust modeling performance of WRF-Chem over the semi-arid and arid regions around the Mediterranean" by Flaounas et al. for publication in Atmospheric Chemistry and Physics The paper presents simulations of the dust loading over three regions of interest (northern Africa, the Mediterranean, and the Middle East) for the six month period of spring and summer 2011. Simulations are made with the WRF-Chem model nudged by meteorology from the ERA-Interim reanalysis. The dust component of the chemistry module was run, and dust was carried as a passive tracer (i.e., no radiative or cloud interactions). Three different dust emissions schemes were considered in order to assess their relative performance at simulating observed dust distributions. Detailed comparisons were made to MODIS and AERONET AOD and aircraft lidar extinction profiles.*

*Each of the three dust emission schemes were run with different scaling factors in order to investigate bias relative to observations. The general conclusion is that none of the emission schemes tried can be optimally configured to provide the best performance over all three regions of interest. There is a secondary conclusion that suggests that simulated surface mass concentrations for the default configurations are excessively large, which can be somewhat ameliorated by including smaller sized dust particles in the simulation (which have lower mass but greater extinction efficiency). My recommendation is that the paper needs major revision. I basically have two major criticisms to the work presented.*

We are thankful to the Reviewer who raised interesting points that helped us improve our manuscript, especially section 4.2. All queries are answered below.

*The first is that the analysis is far too simplistic as presented. Because you are using fixed meteorology and there is no feedback between the dust load and the atmosphere then there is really no point in performing 4 simulations for each of the emission schemes. Of course they have the same correlation coefficient. You are only running dust and no fancy microphysics and the processes simulated are linear (aren't they? I think they are in most models). The rescaling to different tuning coefficients could have been done a posteriori and essentially the same results derived. The analysis could be increased in complexity one notch by doing a regression to find the optical tuning coefficient for each of the source schemes. This of course would not resolve the regional disparities. So maybe a next level of complexity would be to run the simulation turning on and off individual source regions (say, separate eastern from western North Africa an separate from Middle Eastern sources) and then you might find an optimal set of scaling coefficients for each region and each emission scheme (actually, this would require I think only 3 runs for each scheme... all sources, eastern Africa sources, and Middle East sources only, with the western African sources, for example, recovered by differencing the other simulations). This works because no feedback and processes are linear. This would yield some better estimate of the emissions needed in your model to make things work. (As an aside I see that the G01 and MB95 schemes yield very similar results, and they differ from the S04. You don't really get into why this is the case, and that is perhaps interesting.)*

In our analysis we assess the model capacity in simulating AOD in different key regions after applying tuning coefficients. In some extent we do provide an insight into how much tuning should be applied to dust source regions such as North Africa and the Arabian Peninsula. However, our main objective is to test the modeled AOD sensitivity to the use of different emission schemes. Finding an adequate tuning coefficient to reach best results when comparing model to observations is a secondary objective.

We agree with the Reviewer that there is a linear effect when tuning dust sources, and that we could diagnostically reach the same results using fewer simulations. To deepen our analysis, we could perform more sensitivity tests as suggested by the Reviewer. However, in this study we choose to keep our analysis focused on the intercomparison of the model original dust schemes. In the introduction, we are now clearer on our objective priorities. In our conclusion, we also make reference to different approaches that could be undertaken in order to improve model performance as we also highlight the fact that applying coefficients to emissions is only an empirical way to tune model performance.

The schemes parametrisation, as implemented in WRF, do not follow exactly the original publications of G01 and MB95 or S04 on which they were based. To give more emphasis in this issue, we divided Section 2 in three subsections instead of two. Now section 2.2 presents the chemistry component of WRF-Chem explaining in more detail the dust schemes functionality. Finally, to be more precise, we have changed the abbreviations of the schemes into GOCART, AFWA and UoC (University of Cologne). The scheme names are now the same as used in WRF.

Concerning similar performance of GOCART and AFWA we added the following:

*"The modeled AODs vary between the simulations, with the GOCART and the AFWA schemes yielding higher values compared to the UoC scheme. Both GOCART and AFWA simulations seem to produce similar spatial patterns of the dust transport episode and AODs. Since meteorology is identical to all three simulations, the similarity is caused by the AFWA and GOCART emission schemes. Indeed, the same tuning coefficients lead to a similar AOD bias (e.g. Fig. 3) and fairly close correlation coefficients (e.g. Figs 4 and 7). A plausible explanation could reside in the fact that both schemes share the same parametrisation for dry soil threshold friction velocity and that both simulations use soil erodibility to scale dust emission fluxes."*

*The second is that the entire discussion of the dust particle size distribution Section 4.2 is too simplistic. Wouldn't the Fennec observations have some bearing on this discussion? Aren't there are in situ observations of dust mass to compare directly to? The problem I have with this entire line of analysis is that the authors never say how the dust particle size distribution is emitted at the source. For example, the Ginoux et al. 2001 prescription provides one example of an initial particle size distribution. Is that what is being used here? How about MB95, which does not provide explicitly the simulated particle size distribution at emission? And for S04? So some assumption is made about the initial particle size distribution, and again, couldn't the Fennec observations be used to evaluate that? What is not clear is why simply switching from 5 to 8 size bins reduces the simulated dust mass. Obviously you also changed the initial particle size distribution. How? Couldn't you have changed the distribution using the 5 bins to achieve a similar result? All that is described here is what the bin effective radius is, not what the size distribution actually looks like. There's nothing mysterious here about this: if you change the size distribution you change the mass extinction efficiency. Actually, that would be a useful thing to show, how that is different among the simulations.*

The eight bin simulation has been solely performed with the GOCART scheme. The implementation of eight bins to the other schemes was found to be a delicate issue and hence we introduce this simulation only to the discussion part of the article in order to highlight the relationship between dust mass and AOD. The Reviewer correctly points to our discussion being simplistic. We have now added information on the implementation of the eight bins to WRF-Chem.

In fact, the GOCART scheme (based on Ginoux, 2001) assumes that clay accounts for the 10% of the mass of silt, while the silt mass fraction is equally distributed between the silt dust bins. Therefore, the GOCART scheme provides by default a size distribution of equal mass fraction (0.25) for each silt bin, i.e. for the four out of five bins, while the bin of smallest radius (0.73 μm) is given a mass fraction of 0.1 (this forms a sum of mass fractions equal to 1.1).

In the 8-bin simulation presented in the initial manuscript, we applied a rather unrealistic mass fraction distribution. We have now rewritten section 4.2 as we performed two additional sensitivity tests and focused our discussion on mass fraction and mass extinction efficiency. In the first sensitivity test, we followed the assumptions made in the GOCART implementation (mass fraction summing up to 1.1) and applied a mass fraction of 0.025 for each of the first four size bins that we consider to be clay (radii < 1 μm) and 0.25 for the other four bins, considered to be silt (radii > 1 μm). A tuning coefficient of 0.5 has been considered. In the second sensitivity test we modified the rather empirical mass fraction distribution from GOCART to the one

calculated using Kok (2011; their equation 6). Results of the 8-bin simulations still present a "linear effect", when compared to the original 5-bin simulation, but we found that this stimulates an interesting discussion. Section 4.2 has been rewritten.

Kok, J. F.: A scaling theory for the size distribution of emitted dust aerosols suggests climate models underestimate the size of the global dust cycle, Proceedings of the National Academy of Sciences, 108(3), 1016-1021, 2011.

**Minor points:**
*Page 1: Line 25: Should read "however fails to capture..."*

Done

*Line 38: Should read "... the Arabian Peninsula has annual dust emission about one fifth as large as the emissions from North Africa..."*

Done

*Page 2: Line 22: Instead of "resides to" use "relies on"*

Done

*Page 4: Line 22: Why do you call the first scheme G01? You are not using the emission equation from Ginoux et al. 2001, but rather the equation from Gillette & Passi 1988. You are using the Ginoux source map, but you are using that after all for all the simulations. As an aside, could you clarify what resolution of the Ginoux map you are using? Ginoux et al. 2001 describes a 1 degree source map, but he has separately provided a version of that map at 0.25 degree.*

Section 2.2 describes in more detail the schemes functionality. Here we use the 1 degree dataset of Ginoux (2001). This is now stated in the text.

*Page 5: The entire first paragraph is confusing to me. I just don't understand the distinction you are making here between S04 and the G01 and MB95 sources. Is S04 using some other source of sub-grid scale information prior to using the Ginoux map? Otherwise, why isn't this exactly the same as the other two? Fundamentally, don't you use the grid cell resolved winds with some flux equation, and then scale the resulting flux by the efficiency of the grid cell for emissions? Is there some difference in thinking of the Ginoux map as being an efficiency factor versus the fraction of the grid cell available for emissions? Unless there is some other source of sub-grid scale information I don't see what the difference is.*

The AFWA and GOCART schemes use the Ginoux erodibility map to scale emissions, while the UoC scheme does not. In the UoC scheme, the Ginoux erodibility map only serves to define the grid points where dust emission can occur. This point has been clarified in section 2.2.

*Line 31: The OMI data is available nearly time-coincident with the Aqua overpass, but offset in time from the Terra overpass. How do you cope with that in driving this comparison of the model to MODIS data?*

Terra and Aqua satellites cross equator at 10:30 and 13:30 UTC, respectively. To be consistent we compare satellite observations with the model outputs at 12:00 UTC. We now mention this in section 3.1.

*Page 7: Line 7: I don't see how G01-0.75 overestimates the Arabian peninsula AOD in Figure 3d. Line 9: How do you conclude that G01-0.5 produces the most realistic AOD over the Arabian peninsula when it looks more like G01-0.75 does (more white space)?*

Indeed, this was a typo mistake. Lines 7-9 are now corrected:

"....In the Arabian Peninsula, Sim_GOCART-1 tends to overestimate AOD over the southeastern part of the region, compared to the AOD over the northern side. This is consistent with the higher fraction of erodible surface in the south of the Arabian Peninsula, as shown in Fig. 1. Sim_GOCART-0.75 also appears to produce the most realistic AODs in that region...."

*Line 18: You refer to a second scaling. See my comment above about S04 scheme. I guess I'm confused about how the Ginoux map is being applied.*

This is now clarified in section 2.2.

*Page 9: Line 36: I find this conclusion about the background AOD confusing. I could argue that the simulated AOD is less than the background from AERONET at Crete, but it is not clearly the case for Lampedusa. But AERONET shows total column AOD, and you are only simulating the dust component. So I don't buy anything you are saying about the background AOD level. Am I missing something here? Do you have some reason to think that the AERONET AOD shown is due entirely to dust?*

We agree with the Reviewer that this point needs clarification. As stated in section 2.3, the AOD from MODIS has been filtered in order to be representative of dust AOD:

*"Following the same approach as in Flaounas et al. (2015) the MODIS AOD dataset was filtered so that model evaluation is performed only for grid points and days for which dust is present. For this reason, we took into account only AOD values when AE is lower than 0.7 and AI is greater than 1."*

While this is rather difficult for AERONET, where Aerosol Index is not available, we compared the AOD distributions from MODIS (filtered using the criteria AE<0.7 and AI>1 and interpolated to the AERONET locations of Lampedusa and Crete) with the AOD distributions from the two AERONET stations. Results are shown in the figure below (black for MODIS and red for AERONET). Comparison has been done for the AERONET retrievals around 12:00 UTC and only for the days when AERONET and MODIS measurements are available. Although a six month period is rather small to draw robust conclusions, the datasets agree fairly well on a median AOD value of approximately 0.2. In the manuscript we are now clearer on this issue discussing the limitations in our comparison between model and observations:

*"Model comparison with AERONET presents some limitations. While the model calculates only dust related AOD, the AERONET measurements may be also representative of other particulate matter (e.g. sea salt). To gain more confidence in that the 0.2 value of AOD background in AERONET is due to dust, we compared the MODIS AOD retrievals with the AERONET measurements. MODIS measurements have been filtered using the criteria AE<0.7 and AI>1 in order to be representative of dust and have been interpolated to the locations of the AERONET stations at Lampedusa and Crete. The AOD median from MODIS (~0.2) at these locations has been indeed found to be close to the AERONET median."*

[Figure]

*Page 10: Line 23: There is no analysis presented which supports the assertion that the higher altitude part of the dust AOD in the simulation results from long-range transport as opposed emission processes. My experience, admittedly with global models, is that a deep boundary layer develops over Saharan Africa, so that dust may be mixed quite. So what is the PBL height in these simulations and along these profiles?*

It is true that the PBL is quite deep over the Sahara, however the Saharan atmospheric boundary layer (SABL) is not fully mixed until late in the day (usually around 1800 LT, just before sunset), as opposed to what is frequently experienced in the mid-latitude where the PBL is fully developed around noon. First-hand experience gained during AMMA and FENNEC shows that around mid-day, what is observed generally over the Sahara is a developing convective mixing layer (~3 km deep) and a residual layer on top of it (from 3 km to 6 km agl). [The convective mixing layer top reaches that of the residual layer near the end of the day.] Dropsonde measurements for instance made during AMMA and FENNEC around mid-day clearly show distinct temperature inversion associated with the top of each of these layers.

In these conditions, the dust loads in the lower half of the SABL, are generally representative of local emissions. In the upper part of the SABL, dust composition in the Western Sahara is dominated by long-range transport and is characterized by a mixture of dust from many remote sources. This is now more clearly explained in the revised version of the manuscript. We also refer the readers to the recent paper by Chaboureau et al. (2016 ACP) on this matter.

*Figure 10: Use a consistent date labeling for each of sub figures (see title at top of each figure).*

Done

---

## Author Comment (AC2) · 3 Sep 2016

*This paper reports on testing the performance of a regional dust-atmospheric modeling system. The study aims at optimizing the WRF-Chem model performance with added dust aerosol component in order to be capable to operationally forecast of dust transport over the eastern Mediterranean. The presented model is another one in the family of dust prognostic systems which development follows the interest of community to better predict dust process and its various impacts.*

*The authors successfully performed a series of tests to understand the performance of three used emission schemes, and to tune the model to achieve the optimal accuracy in different regions of the model domain. I recommend this paper to be accepted for publication after the authors consider suggestions and revisions as listed below:*

We would like to thank the Reviewer for his/her constructive comments that helped us improve the manuscript. All suggestions have been taken into account and the paper has been changed accordingly.

*Page 1 line 28: Tuning the model performance by applying a coefficient to dust emissions I agree this is the most straightforward way to vary the intensity of emissions and accept it as one of ways to tune the model. However, by this approach only a linear change of values every time everywhere is done. There are other possibilities as well such as e.g. modifying values of the threshold surface wind or friction velocity, aeolian surface roughness length, etc. Please discuss more this aspect and other possible ways for tunning.*

Our main objective is to test the model sensitivity to different emission schemes. Tuning was used here only as an empirical modification in order to adjust model outputs into a realistic level. We have rewritten parts of the introduction in order to be clearer that tuning *per se* is a secondary objective. In addition, we now discuss other ways of improving model performance in the conclusion. Finally, we performed additional sensitivity tests enriching the discussion in section 4.2 where mass fraction of dust bins has been modified for the 8-bin simulation (only for the GOCART simulation). We added the following in the conclusion:

*"Empirical tuning of dust emissions has no physical basis and corresponds to a model adjustment that is valid for the specific model set-up (e.g. grid spacing, number of vertical levels, physical parametrisation). In fact, applying tuning only modifies linearly the model performance. Optimization of dust emissions would demand modifications of the parametrization (e.g. change the thresholds of surface and friction wind speeds) or the relevant surface fields (e.g soil erodibility). Such modifications focus on modelling assumptions and thus provide a more physics oriented optimization of the model performance. Given the differences in the physical assumptions of the dust schemes, such sensitivity tests could only focus however on specific parametrisations yielding non-linear effects on the results. Therefore, future work will be concentrated to further test the model sensitivity to realistically reproduce dust transport events using both eight dust size bins and finer model resolutions. Furthermore, we will concentrate on the climatology of dust transport over the Mediterranean by performing long term simulations also aiming at investigating the aerosols direct and indirect effect."*

*Page 3 line 138: we nudged wind, temperature and water vapour at each grid point to the ERA-I reanalysis: The authors claim that one of their objectives is ...to establish an empirically tuned dust forecasting model for the effective forecast of dust transport... By using nudging, operational features of the model could be contaminated. Once used, why nudging is not applied only to wind as the most critical parameter for emission? My general concern is that frequent nudging as applied in the experiment could affect the thermodynamic features of the atmosphere with unknown consequences. Please discuss possible impact of nudging to the operatibility of the model and eventual affecting the model thermodynamic balance.*

Our objective is to find an optimal dust emission configuration for the purposes of operational dust forecasting. For this reason our model results are compared to satellite and ground observations of AOD. Valid conclusions from such comparison require that the modeled AOD uncertainties are entirely -at best- or in majority related to the WRF emission schemes and not e.g. to uncertainties related to meteorology. To this end, using nudging we

introduce to the model additional tendencies on wind, temperature and water vapor. These tendencies guide the model outputs in order not to diverge from the reanalysis. The use of nudging is hence a convenient trade-off that bounds model performance to the reanalysis (i.e. reduces feedback of modeled physical processes to the outputs) but relaxes the model outputs towards the "more realistic" reanalysis. Nudging would indeed jeopardize the robustness of our conclusion if for instance we were testing the model sensitivity to dust direct and indirect effect. However, here we do not treat explicitly such issues. We are now clearer in our motivation to use nudging in the introduction.

*"In particular, it was found that applying nudging reduces the model 10-meter wind speed absolute bias over North Africa by approximately 35%, while it also allows for a better subjective agreement between the observed and modelled synoptic-scale patterns associated with dust transport. Furthermore, in long simulations of more than a few days, nudging is beneficial in reducing uncertainties in the atmospheric circulation due to the model internal variability. Our choice to nudge is thus based on achieving realistic seasonal atmospheric circulation over the domain which is particularly important for dust emissions. Since nudging introduces additional tendencies to the model for wind, temperature and water vapour, it would affect our results if only we compared simulations that treat dust direct and indirect effects. However, here dust is treated as a passive tracer."*

**General:**
*The presented extensive verification is certainly a good guidance how to select model setup based on more reliable emission options. However, since the authors' intention is to have a well tested model to be used for forecasting purposes, I strongly suggest that they select one of major dust storms during the considered experiment period and present a more close-up view so that a reader could get a better feeling on the model capability to successfully predict particular dust events.*

We agree with the Reviewer. We now comment and include a new figure in section 3.2. This figure presents a dust transport episode that took place in July 23 over the eastern Mediterranean. In the end of the section, we qualitatively compare model to observations.